# PBP Target Profiling by $\beta$-Lactam and $\beta$-Lactamase Inhibitors in Intact *Pseudomonas aeruginosa*: Effects of the Intrinsic and Acquired Resistance Determinants on the Periplasmic Drug Availability

Maria Montaner,[a] Silvia Lopez-Argüello,[a] ⓘ Antonio Oliver,[a,b] ⓘ Bartolome Moya[a,b]

aServicio de Microbiología y Unidad de Investigación, Hospital Universitario Son Espases, Health Research Institute of the Balearic Islands (IdISBa), Palma, Spain
bCIBER in Infectious Diseases (CIBERINFEC), Madrid, Spain

Maria Montaner and Silvia Lopez-Argüello contributed equally to this work. Author order was determined by drawing straws.

**ABSTRACT**   The lack of effective treatment options against *Pseudomonas aeruginosa* is one of the main contributors to the silent pandemic. Many antibiotics are ineffective against resistant isolates due to poor target site penetration, efflux, or $\beta$-lactamase hydrolysis. Critical insights to design optimized antimicrobial therapies and support translational drug development are needed. In the present work, we analyzed the periplasmic drug uptake and binding to PBPs of 11 structurally different $\beta$-lactams and 4 $\beta$-lactamase inhibitors (BLIs) in *P. aeruginosa* PAO1. The contribution of the most prevalent $\beta$-lactam resistance mechanisms to MIC and periplasmic target attainment was also assessed. Bacterial cultures (6.5 $\log_{10}$ CFU/mL) were exposed to $1/2\times$ PAO1 MIC of each antibiotic for 30 min. Unbound PBPs were labeled with Bocillin FL and analyzed using a FluorImager. Imipenem extensively inactivated all targets. Cephalosporins preferentially targeted PBP1a and PBP3. Aztreonam and amdinocillin bound exclusively to PBP3 and to PBP2 and PBP4, respectively. Penicillins bound preferentially to PBP1a, PBP1b, and PBP3. BLIs displayed poor PBP occupancy. Inactivation of *oprD* elicited a notable reduction of imipenem target attainment, and it was to a lesser extent in the other carbapenems. Improved PBP occupancy was observed for the main targets of the widely used antipseudomonal penicillins, cephalosporins, meropenem, aztreonam, and amdinocillin upon *oprM* inactivation, in line with MIC changes. AmpC constitutive hyperexpression caused a substantial PBP occupancy reduction for the penicillins, cephalosporins, and aztreonam. Data obtained in this work will support the rational design of optimized $\beta$-lactam-based combination therapies against resistant *P. aeruginosa* infections.

**IMPORTANCE**   The growing problem of antibiotic resistance in Gram-negative pathogens is linked to three key aspects, (i) the progressive worldwide epidemic spread of multidrug-resistant (MDR), extensively drug-resistant (XDR), and pandrug-resistant (PDR) Gram-negative strains, (ii) a decrease in the number of effective new antibiotics against multiresistant isolates, and (iii) the lack of mechanistically informed combinations and dosing strategies. Our combined efforts should focus not only on the development of new antimicrobial agents but the adequate administration of these in combination with other agents currently available in the clinic. Our work determined the effectiveness of these compounds in the clinically relevant bacteria *Pseudomonas aeruginosa* at the molecular level, assessing the net influx rate and their ability to access their targets and achieve bacterial killing without generating resistance. The data generated in this work will be helpful for translational drug development.

**KEYWORDS**   penicillin-binding proteins (PBP), intact cells, *Pseudomonas aeruginosa*, outer membrane penetration, bacterial permeability, $\beta$-lactams, $\beta$-lactam resistance, antimicrobial resistance

Address correspondence to Bartolome Moya, bartolome.moya@ssib.es.

The authors declare no conflict of interest.

**M**ultidrug-resistant (MDR) *Pseudomonas aeruginosa* has been a relentless threat during the last decades and one of the main drivers of the so-called silent pandemic. This microorganism has become one of the leading causes of nosocomial infections with higher rates of poor clinical outcome and mortality (1). There is a severe shortage of new antibiotics entering late phases of clinical development to treat infections caused by *P. aeruginosa*, frequently resistant to all available monotherapies (2, 3).

β-Lactams are the most widely used class of antibacterial agents owing to their broad spectrum and proven safety profiles. Clinical trials in the 1970s and 1980s studied combination therapies with promising results for empirical nonoptimized double β-lactam therapies (4). Most prevalent β-lactam resistance mechanisms in *P. aeruginosa* PAO1 clinical isolates include β-lactamase derepression (AmpC; penicillins and cephalosporins affected to different degrees), which causes a reduction of the effective concentration of labile drugs in the periplasm and activation or derepression of active efflux systems (MexAB-OprM; penicillins, cephalosporins, monobactams, and meropenem), decreasing the periplasmic concentration of the affected drugs and decreasing outer membrane (OM) expression of the specific porin OprD (imipenem and other carbapenems to a lesser degree). MDR strains arise from interplay of these resistance determinants, among others (5).

With the current increase of resistant *P. aeruginosa* isolates, new treatment options such as ceftazidime-avibactam, ceftolozane-tazobactam, meropenem-vaborbactam, and cefiderocol have become available (6), adding new resources into the armamentarium to counteract severe infections due to MDR organisms. Despite important progress, pharmacokinetic/pharmacodynamic (PK/PD) optimization of dosage regimens and treatment duration in critically ill patients requires further study, taking into account resistance selection as a major endpoint. Furthermore, several studies have described the development of resistance to new compounds during or after treatment (7, 8).

β-Lactams bind to various penicillin-binding proteins (PBPs; with highly conserved active sites) with different affinities forming stable covalent complexes (9). Structurally different β-lactams with the same PBP-binding profiles and selectivity (50% inhibitory concentration [$IC_{50}$]) achieve different MICs and rates of bacterial killing (6, 10). MIC summarizes bacterial killing and resistance emergence during 18 h of bacterial growth. Most often, dose and drug selection are based on this static *in vitro* parameter. Although MICs combined with dynamic pharmacokinetic parameters are often useful predictors of the drug-microorganism interaction outcome, many PK/PD inconsistencies have been observed (11). The effectiveness of a β-lactam depends on its rate of penetration and, ultimately, on target attainment (12, 13). The outer membrane architecture of Gram-negatives represents a major barrier to the target site penetration of many antibiotics (14). Moreover, *P. aeruginosa* possesses an even less permeable OM (15).

Isolated PBP-binding assays provide very valuable information with some limitations. PBPs are extensively modified and studied outside their natural environment in bacterial cells (periplasm) (16, 17). Furthermore, in live cells, β-lactam molecules must penetrate the outer membrane and compete with β-lactamases and efflux to bind to different PBPs in the periplasmic space. Moreover, inducible chromosomal resistance can arise in the presence of some β-lactams (inducers) (16, 18). Measuring the OM permeability and the differential target binding of β-lactams in the presence or absence of resistance mechanisms is fundamental to developing optimized therapies and combinations to treat infections caused by MDR *P. aeruginosa* (19, 20).

Besides imaging studies based on cell morphology caused by the inhibition of specific PBPs, only a small number of recent publications have studied PBP binding in live *Escherichia coli* and *P. aeruginosa*, highlighting the lack of correlation between β-lactams structural similarity, PBP inhibition profile, and MIC (10, 21).

Even though β-lactams still represent the most widely used class of antibiotics nowadays, fundamental gaps remain in understanding their mechanism of action and target site penetration and binding (22, 23). Whole-cell assays are very much needed to provide mechanistic data to inform and cross-validate contemporary molecular approaches (18, 22, 23).

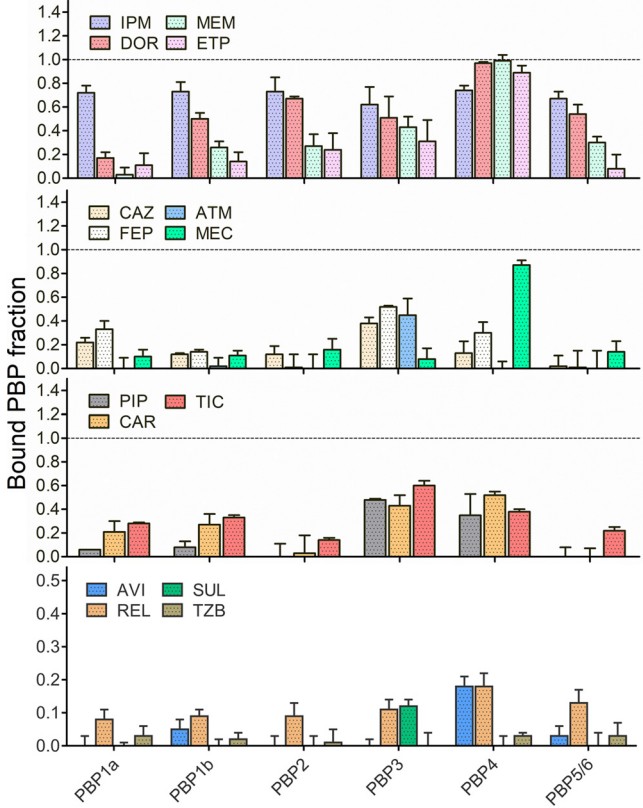

**FIG 1** Bound PBP fraction by β-lactams and BLIs in whole-cell *P. aeruginosa* PAO1. The columns represent the mean values and standard deviation of bound PBP fraction compared to the untreated control obtained from at least three independent experiments. Dashed lines represent 100% of PBP binding. Please see Table 2 for drug abbreviations.

In this work, we studied the PBP binding in *P. aeruginosa* PAO1 intact cells. We sought to correlate the observed MIC differences with the differential intact cell PBP occupancy of compounds with similar PBP IC$_{50}$s (from isolated membrane assays). We created the first intact cell PBP occupancy data set for 11 chemically diverse β-lactams and 4 β-lactamase inhibitors in *P. aeruginosa* PAO1. Furthermore, to validate our results, we determined the binding occupancies in isogenic PAO1 mutants lacking *oprD* (outer membrane permeability), *oprM* (MexAB and MexXY efflux pump inactivation), *ampC* (AmpC inactivation), and *dacB* (AmpC hyperexpression). The data obtained in the present study will inform a new generation of mathematical models to better understand the mechanisms of antibiotic target site penetration, synergy, and resistance. This information will allow us to rationally design and optimize promising drug combination dosing strategies, which will be far superior to the empirical, nonoptimized regimens (11).

## RESULTS

**PBP binding in whole *P. aeruginosa* PAO1 cells.** To better understand differences in MIC values and killing rates of compounds sharing similar lysed-cell PBP occupancy patterns, we determined the extent of PBP binding of 11 structurally different β-lactams (1/2× PAO1 MIC) and 4 β-lactamase inhibitors (4 mg/L) in mid-exponentially growing (6.5 log$_{10}$ CFU/mL) PAO1 intact cells (Fig. 1). Static concentration time-kill curves revealed no significant bacterial killing (which would compromise OM integrity and expose PBPs to the drugs) for any of the drugs or strains studied after 30 min incubation (see Fig. S1 in the supplemental material). Neither significant dissociation of the PBP-drug complex (during membrane isolation) nor displacement of the drug by Bocillin FL incubation was observed with our experimental approach (Table S1).

Carbapenems showed the most extensive binding in whole cells, showing the highest occupancies for PBP2, PBP4, and PBP5/6 (Fig. 1). Compared to the lysed cell membrane

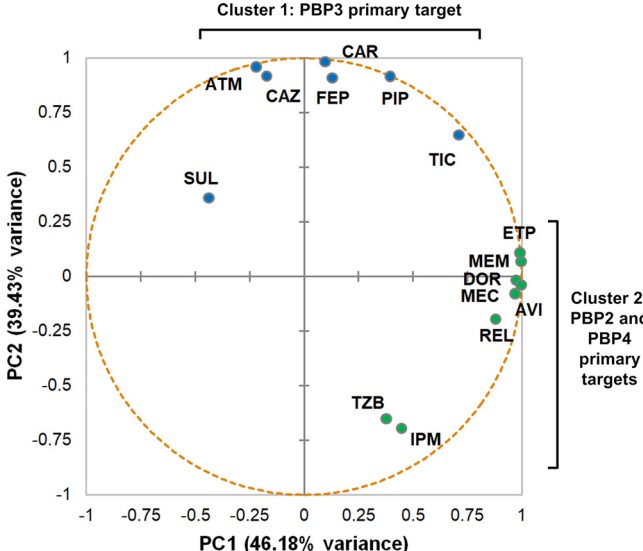

**FIG 2** Principal-component analysis (PCA) of the log-transformed PBP fraction of unbound data of the 15 tested compounds in whole-cell *P. aeruginosa* PAO1. Compounds are clustered according to their positions on the first and second eigenvector, explaining 85.61% of the total variance in the intact cell assay. Please see Table 2 for drug abbreviations.

preparation $IC_{50}$ binding data, the main difference was a higher PBP5/6 occupancy (Table S2). Among this drug class, imipenem demonstrated the most extensive binding, with an ~70% PBP attainment for all the targets except PBP3 after 30 min. Doripenem displayed a similar binding profile for PBP2, PBP3, PBP4, and PBP5/6 in whole cells, but with a certainly lower PBP1a and PBP1b binding. The PBP selectivity matched the lysed membrane preparation with a superior PBP5/6 binding. Meropenem yielded lower overall binding in whole cells, especially for PBP1a, PBP1b, and PBP5/6 compared to imipenem, but with a higher extent of PBP4 binding (99%). However, as previously reported, binding $IC_{50}$s and profiles of these two drugs were comparable in lysed cell assays. Ertapenem, on the other hand, achieved the lowest intact cell PBP occupancy within the carbapenems class, but in lysed cells, $IC_{50}$ values were similar to doripenem. Only PBP4 was steadily inhibited by ertapenem, while PBP1a, PBP1b, PBP2, and PBP5/6 achieved barely 8% to 31% of PBP occupancy.

Cephalosporins were selective for PBP1a and PBP3 in whole cells, though to a lesser extent for PBP occupancy (20% to 50%) than the carbapenems. However, $IC_{50}$ values for these PBPs were comparable in lysed cell assays. Aztreonam showed consistent and selective PBP3 binding (~50% PBP inactivation) in whole and lysed cells. On the other hand, amdinocillin displayed moderate PBP2 binding and surprisingly high PBP4 occupancy as opposed to the exclusive PBP2 binding observed in lysed cells.

Penicillins bound preferentially to PBP3 and PBP4 with modest PBP1a and PBP1b binding, whereas PBP2 and PBP5/6 binding was barely observed. Two milligrams per liter of piperacillin elicited 50% of PBP3 binding in whole cells; however, a 40-fold-lower concentration was enough to half-maximally inhibit this PBP in lysed cells. At 4 mg/L, BLIs showed a very modest PBP binding (<20%) in whole cells. Tazobactam showed no significant binding, sulbactam bound PBP3, and avibactam and relebactam were selective for PBP1b and PBP4.

A principal-component analysis (PCA) of the log-transformed bound PBP fraction (Fig. 2) showed that the first two eigenvectors explained 39.43% and 46.18% of the variance in PAO1 whole-cell PBP binding, respectively, predicting 85.61% of the total variance. Compounds were grouped into two general clusters according to their primary target. The first cluster contained β-lactams that primarily targeted PBP3, penicillins, cephalosporins, aztreonam, and sulbactam. The second cluster was comprised of compounds that mainly inactivated PBP2 and PBP4, carbapenems, amdinocillin, and the BLIs avibactam, relebactam, and tazobactam. All

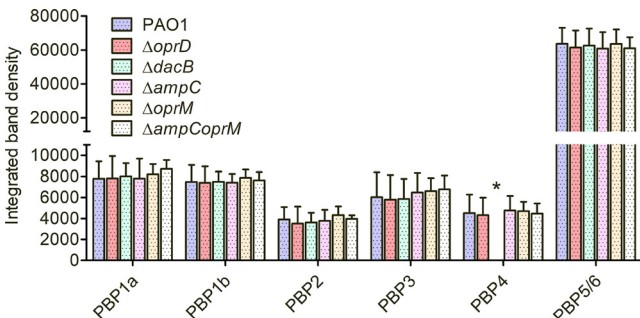

**FIG 3** Untreated membrane preparations were labeled with Bocillin FL to determine changes in band intensities across the isogenic PAO1 mutants used in this study. The columns show the mean peak integration values and standard deviation of PBP signal obtained from at least three independent experiments. Strains studied include PAO1, OprD porin (Δ*oprD*), PBP4 (Δ*dacB*), AmpC β-lactamase (Δ*ampC*), OprM efflux component (Δ*oprM*), and double Δ*ampCoprM* isogenic PAO1 knockout mutants. One-way ANOVA with *post hoc* Tukey's multiple-comparison tests were run, but no statistically significant differences were found. *, PBP4 band absent in Δ*dacB* knockout mutant.

of the studied compounds except amdinocillin showed equivalent PBP selectivity to lysed cell PBP binding data.

**Differential band intensities of PBPs in isogenic *P. aeruginosa* strains.** To assess if the gene knockout could alter the band intensity of a given PBP in the different *P. aeruginosa* isogenic mutants compared to the parent strain PAO1, the integrated band density values were determined for each PBP in untreated controls (Fig. 3). No significant differences in any of the PBP signals were observed. Besides the apparent lack of PBP4 in the PAOΔ*dacB* mutant, no significant signal alteration was observed for the rest of the PBPs.

***P. aeruginosa* β-lactam resistance determinants effect on whole-cell PBP binding.** To better understand the effect of intrinsic resistance determinants (basal efflux pumps and β-lactamase expression) on the MIC and periplasmic target binding, the PBP binding was studied in PAΔ*ampC*, PAΔ*oprM*, and PAΔ*ampCoprM* intact cells and compared to parental strain PAO1 (Fig. S2). Furthermore, to assess if our employed method would be sensitive to changes in periplasmic drug accumulation and target binding in the context of acquired resistance mutations, the intact cell PBP occupancy was also assessed in OprD-inactivated and AmpC-hyperexpressing isogenic mutants (PAΔ*oprD*, PAΔ*dacB*) (Fig. S2).

The bound PBP fraction in intact cells and MICs (for reference) of every strain studied in this work are shown in Table 1. OprM inactivation caused the greatest MIC reduction for meropenem together with a significant increase in target binding (PBP1a, PBP1b, and PBP2). Furthermore, *ampC and oprM* double inactivation caused an additional increase in meropenem and ertapenem PBP occupancy for PBP1a, PBP1b, PBP2, and PBP3, along with an ~15-fold MIC reduction. Regarding chromosomally acquired mutations, carbapenem PBP occupancies were strongly influenced by the inactivation of the substrate-specific outer membrane porin OprD, especially imipenem (1.6- to 3.2-fold reduction of PBP occupancies) and, to a lesser extent, doripenem, meropenem, and ertapenem, in accordance with MIC changes (Table 1). AmpC hyperexpression (Δ*dacB*) did not elicit substantial differences in target binding for any of the carbapenems. Although imipenem MIC diminished 8-fold, no significant increase of target binding was observed upon *ampC* inactivation.

Inactivation of *ampC* did not generate a significant increase in PBP target binding among cephalosporins. However, constitutive hyperexpression of the β-lactamase significantly reduced ceftazidime (PBP1a, PBP1b, and PBP3) and cefepime (PBP1a and PBP1b) PBP occupancies (1.21- to 3.2-fold). The OprM-defective mutant also showed an increase in cefepime PBP binding for all the receptors except PBP2 (1.4-fold average), which was more prominent when *ampC* was additionally inactivated (2.6-fold average). Ceftazidime, on the other hand, showed smaller increases in PBP occupancies along with the MIC reduction.

Aztreonam PBP occupancy was mostly affected by OprM and dual AmpC-OprM inactivation, determining binding increases (1.5- to 36-fold) for PBP1b, PBP3, and PBP4 with the subsequent 16-fold MIC reduction in both cases. Amdinocillin binding increases on the OprM,

**TABLE 1** Bound PBP fraction in isogenic PAO1 *P. aeruginosa* strains

| Drug | MIC or PBP | Bound PBP fraction in strain[a]: | | | | | |
|---|---|---|---|---|---|---|---|
| | | PAO1 | *oprD* | Δ*dacB* | Δ*ampC* | Δ*oprM* | Δ*ampCoprM* |
| IPM | MIC (mg/L) | 1 | 8 | 1 | 0.125 | 1 | 0.25 |
| | PBP1a | 0.72 ± 0.06 | 0.34 ± 0.03 | 0.66 ± 0.07 | 0.81 ± 0.03 | 0.64 ± 0.03 | 0.74 ± 0.02 |
| | PBP1b | 0.73 ± 0.08 | 0.37 ± 0.06 | 0.64 ± 0.08 | 0.82 ± 0.01 | 0.67 ± 0.04 | 0.76 ± 0.07 |
| | PBP2 | 0.73 ± 0.12 | 0.26 ± 0.12 | 0.76 ± 0.02 | 0.71 ± 0.04 | 0.64 ± 0.11 | 0.78 ± 0.03 |
| | PBP3 | 0.62 ± 0.15 | 0.28 ± 0.11 | 0.46 ± 0.05 | 0.68 ± 0.10 | 0.51 ± 0.14 | 0.43 ± 0.11 |
| | PBP4 | 0.74 ± 0.04 | 0.51 ± 0.14 | NA | 0.72 ± 0.08 | 0.52 ± 0.11 | 0.80 ± 0.05 |
| | PBP5/6 | 0.67 ± 0.06 | 0.41 ± 0.12 | 0.84 ± 0.02 | 0.50 ± 0.05 | 0.59 ± 0.08 | 0.38 ± 0.11 |
| DOR | MIC (mg/L) | 0.5 | 1 | 0.5 | 0.125 | 0.125 | 0.064 |
| | PBP1a | 0.17 ± 0.05 | 0.00 ± 0.08 | 0.13 ± 0.05 | 0.10 ± 0.01 | 0.15 ± 0.03 | 0.21 ± 0.11 |
| | PBP1b | 0.50 ± 0.05 | 0.22 ± 0.07 | 0.47 ± 0.13 | 0.49 ± 0.02 | 0.43 ± 0.09 | 0.69 ± 0.02 |
| | PBP2 | 0.67 ± 0.02 | 0.33 ± 0.01 | 0.62 ± 0.10 | 0.68 ± 0.10 | 0.62 ± 0.04 | 0.77 ± 0.06 |
| | PBP3 | 0.51 ± 0.18 | 0.47 ± 0.00 | 0.55 ± 0.06 | 0.55 ± 0.01 | 0.51 ± 0.04 | 0.63 ± 0.01 |
| | PBP4 | 0.97 ± 0.00 | 0.95 ± 0.00 | NA | 0.97 ± 0.07 | 0.98 ± 0.01 | 0.98 ± 0.03 |
| | PBP5/6 | 0.54 ± 0.08 | 0.06 ± 0.08 | 0.52 ± 0.12 | 0.00 ± 0.05 | 0.77 ± 0.02 | 0.69 ± 0.14 |
| MEM | MIC (mg/L) | 0.5 | 1 | 1 | 0.125 | 0.032 | 0.032 |
| | PBP1a | 0.03 ± 0.06 | 0.01 ± 0.02 | 0.03 ± 0.11 | 0.00 ± 0.10 | 0.22 ± 0.01 | 0.74 ± 0.01 |
| | PBP1b | 0.26 ± 0.05 | 0.07 ± 0.01 | 0.16 ± 0.06 | 0.21 ± 0.06 | 0.58 ± 0.03 | 0.69 ± 0.06 |
| | PBP2 | 0.27 ± 0.10 | 0.16 ± 0.07 | 0.22 ± 0.19 | 0.33 ± 0.13 | 0.61 ± 0.13 | 0.66 ± 0.07 |
| | PBP3 | 0.43 ± 0.09 | 0.29 ± 0.10 | 0.44 ± 0.03 | 0.36 ± 0.01 | 0.54 ± 0.03 | 0.59 ± 0.01 |
| | PBP4 | 0.99 ± 0.04 | 0.94 ± 0.06 | NA | 0.95 ± 0.01 | 1.00 ± 0.00 | 0.98 ± 0.03 |
| | PBP5/6 | 0.30 ± 0.05 | 0.00 ± 0.06 | 0.04 ± 0.02 | 0.00 ± 0.05 | 0.46 ± 0.23 | 0.39 ± 0.09 |
| ETP | MIC (mg/L) | 8 | 16 | 16 | 4 | 2 | 0.5 |
| | PBP1a | 0.11 ± 0.10 | 0.17 ± 0.01 | 0.14 ± 0.03 | 0.05 ± 0.20 | 0.17 ± 0.12 | 0.37 ± 0.01 |
| | PBP1b | 0.14 ± 0.08 | 0.21 ± 0.04 | 0.13 ± 0.07 | 0.06 ± 0.25 | 0.29 ± 0.07 | 0.60 ± 0.06 |
| | PBP2 | 0.24 ± 0.14 | 0.25 ± 0.03 | 0.15 ± 0.17 | 0.11 ± 0.12 | 0.30 ± 0.00 | 0.63 ± 0.02 |
| | PBP3 | 0.31 ± 0.18 | 0.29 ± 0.06 | 0.22 ± 0.09 | 0.20 ± 0.21 | 0.41 ± 0.05 | 0.63 ± 0.02 |
| | PBP4 | 0.89 ± 0.06 | 0.68 ± 0.10 | NA | 0.87 ± 0.01 | 0.99 ± 0.01 | 0.98 ± 0.02 |
| | PBP5/6 | 0.08 ± 0.12 | 0.27 ± 0.16 | 0.24 ± 0.09 | 0.08 ± 0.14 | 0.97 ± 0.06 | 0.64 ± 0.18 |
| CAZ | MIC (mg/L) | 1 | 1 | 8 | 0.5 | 0.25 | 0.25 |
| | PBP1a | 0.22 ± 0.04 | 0.21 ± 0.09 | 0.05 ± 0.17 | 0.18 ± 0.06 | 0.40 ± 0.11 | 0.59 ± 0.02 |
| | PBP1b | 0.12 ± 0.01 | 0.06 ± 0.08 | 0.00 ± 0.14 | 0.11 ± 0.16 | 0.14 ± 0.03 | 0.25 ± 0.03 |
| | PBP2 | 0.12 ± 0.17 | 0.00 ± 0.13 | 0.11 ± 0.11 | 0.02 ± 0.24 | 0.05 ± 0.09 | 0.26 ± 0.04 |
| | PBP3 | 0.38 ± 0.05 | 0.47 ± 0.09 | 0.18 ± 0.05 | 0.44 ± 0.08 | 0.55 ± 0.00 | 0.60 ± 0.06 |
| | PBP4 | 0.13 ± 0.10 | 0.15 ± 0.06 | NA | 0.15 ± 0.23 | 0.06 ± 0.01 | 0.30 ± 0.10 |
| | PBP5/6 | 0.02 ± 0.09 | 0.14 ± 0.09 | 0.00 ± 0.07 | 0.03 ± 0.11 | 0.00 ± 0.02 | 0.27 ± 0.17 |
| FEP | MIC (mg/L) | 1 | 1 | 4 | 1.00 | 0.125 | 0.125 |
| | PBP1a | 0.33 ± 0.07 | 0.39 ± 0.15 | 0.20 ± 0.06 | 0.25 ± 0.08 | 0.46 ± 0.09 | 0.56 ± 0.08 |
| | PBP1b | 0.14 ± 0.02 | 0.21 ± 0.12 | 0.05 ± 0.00 | 0.11 ± 0.05 | 0.13 ± 0.09 | 0.27 ± 0.13 |
| | PBP2 | 0.01 ± 0.11 | 0.07 ± 0.08 | 0.05 ± 0.03 | 0.03 ± 0.07 | 0.00 ± 0.07 | 0.06 ± 0.03 |
| | PBP3 | 0.52 ± 0.01 | 0.52 ± 0.04 | 0.51 ± 0.05 | 0.48 ± 0.08 | 0.55 ± 0.05 | 0.61 ± 0.03 |
| | PBP4 | 0.30 ± 0.09 | 0.33 ± 0.09 | NA | 0.31 ± 0.05 | 0.70 ± 0.11 | 0.81 ± 0.01 |
| | PBP5/6 | 0.01 ± 0.04 | 0.24 ± 0.07 | 0.04 ± 0.10 | 0.00 ± 0.07 | 0.21 ± 0.02 | 0.12 ± 0.08 |
| ATM | MIC (mg/L) | 2 | 2 | 8 | 2 | 0.125 | 0.125 |
| | PBP1a | 0.00 ± 0.09 | 0.00 ± 0.06 | 0.01 ± 0.04 | 0.00 ± 0.10 | 0.00 ± 0.08 | 0.09 ± 0.06 |
| | PBP1b | 0.02 ± 0.07 | 0.00 ± 0.03 | 0.03 ± 0.08 | 0.00 ± 0.14 | 0.35 ± 0.02 | 0.45 ± 0.08 |
| | PBP2 | 0.00 ± 0.12 | 0.00 ± 0.19 | 0.00 ± 0.03 | 0.00 ± 0.07 | 0.00 ± 0.12 | 0.00 ± 0.03 |
| | PBP3 | 0.45 ± 0.14 | 0.51 ± 0.03 | 0.25 ± 0.14 | 0.51 ± 0.02 | 0.59 ± 0.14 | 0.58 ± 0.15 |
| | PBP4 | 0.00 ± 0.06 | 0.11 ± 0.07 | NA | 0.240 ± 0.16 | 0.24 ± 0.12 | 0.36 ± 0.05 |
| | PBP5/6 | 0.00 ± 0.11 | 0.08 ± 0.07 | 0.17 ± 0.25 | 0.00 ± 0.16 | 0.00 ± 0.08 | 0.03 ± 0.13 |
| MEC | MIC (mg/L) | 256 | 256 | >256 | 256 | 4 | 4 |
| | PBP1a | 0.10 ± 0.06 | 0.00 ± 0.03 | 0.00 ± 0.10 | 0.00 ± 0.06 | 0.11 ± 0.02 | 0.10 ± 0.03 |
| | PBP1b | 0.11 ± 0.04 | 0.00 ± 0.02 | 0.00 ± 0.12 | 0.30 ± 0.05 | 0.00 ± 0.06 | 0.19 ± 0.01 |
| | PBP2 | 0.16 ± 0.09 | 0.09 ± 0.06 | 0.02 ± 0.07 | 0.00 ± 0.02 | 0.14 ± 0.08 | 0.10 ± 0.00 |
| | PBP3 | 0.08 ± 0.09 | 0.00 ± 0.05 | 0.00 ± 0.12 | 0.00 ± 0.14 | 0.00 ± 0.08 | 0.15 ± 0.11 |
| | PBP4 | 0.87 ± 0.04 | 0.77 ± 0.01 | NA | 0.88 ± 0.00 | 0.55 ± 0.02 | 0.90 ± 0.02 |
| | PBP5/6 | 0.14 ± 0.09 | 0.00 ± 0.12 | 0.00 ± 0.06 | 0.15 ± 0.03 | 0.06 ± 0.14 | 0.00 ± 0.20 |

**TABLE 1** (Continued)

| Drug | MIC or PBP | Bound PBP fraction in strain[a]: | | | | | |
|---|---|---|---|---|---|---|---|
| | | PAO1 | *oprD* | Δ*dacB* | Δ*ampC* | Δ*oprM* | Δ*ampCoprM* |
| PIP | MIC (mg/L) | 4 | 4 | 16 | 2 | 0.25 | 0.125 |
| | PBP1a | 0.06 ± 0.00 | 0.15 ± 0.07 | 0.00 ± 0.10 | 0.00 ± 0.10 | 0.30 ± 0.06 | 0.73 ± 0.03 |
| | PBP1b | 0.08 ± 0.05 | 0.19 ± 0.09 | 0.00 ± 0.10 | 0.00 ± 0.04 | 0.24 ± 0.12 | 0.54 ± 0.06 |
| | PBP2 | 0.00 ± 0.11 | 0.07 ± 0.12 | 0.00 ± 0.07 | 0.00 ± 0.19 | 0.00 ± 0.15 | 0.32 ± 0.16 |
| | PBP3 | 0.48 ± 0.01 | 0.49 ± 0.07 | 0.33 ± 0.07 | 0.45 ± 0.17 | 0.46 ± 0.09 | 0.60 ± 0.25 |
| | PBP4 | 0.35 ± 0.18 | 0.45 ± 0.18 | NA | 0.32 ± 0.19 | 0.91 ± 0.04 | 0.99 ± 0.01 |
| | PBP5/6 | 0.00 ± 0.08 | 0.19 ± 0.23 | 0.00 ± 0.03 | 0.10 ± 0.18 | 0.00 ± 0.18 | 0.21 ± 0.18 |
| CAR | MIC (mg/L) | 64 | 64 | 128 | 64 | 1 | 1 |
| | PBP1a | 0.21 ± 0.09 | 0.27 ± 0.14 | 0.24 ± 0.04 | 0.21 ± 0.13 | 0.65 ± 0.08 | 0.89 ± 0.01 |
| | PBP1b | 0.27 ± 0.09 | 0.32 ± 0.17 | 0.26 ± 0.04 | 0.25 ± 0.09 | 0.66 ± 0.10 | 0.89 ± 0.01 |
| | PBP2 | 0.03 ± 0.05 | 0.11 ± 0.15 | 0.02 ± 0.02 | 0.00 ± 0.08 | 0.05 ± 0.17 | 0.07 ± 0.18 |
| | PBP3 | 0.43 ± 0.09 | 0.46 ± 0.10 | 0.48 ± 0.01 | 0.44 ± 0.05 | 0.54 ± 0.06 | 0.60 ± 0.10 |
| | PBP4 | 0.52 ± 0.03 | 0.54 ± 0.04 | NA | 0.49 ± 0.14 | 0.83 ± 0.03 | 0.96 ± 0.02 |
| | PBP5/6 | 0.00 ± 0.07 | 0.17 ± 0.11 | 0.00 ± 0.10 | 0.01 ± 0.11 | 0.22 ± 0.12 | 0.12 ± 0.24 |
| TIC | MIC (mg/L) | 16 | 16 | 64 | 16 | 0.50 | 0.50 |
| | PBP1a | 0.28 ± 0.01 | 0.2 ± 0.11 | 0.13 ± 0.05 | 0.26 ± 0.02 | 0.60 ± 0.05 | 0.91 ± 0.01 |
| | PBP1b | 0.33 ± 0.02 | 0.28 ± 0.11 | 0.17 ± 0.04 | 0.26 ± 0.04 | 0.60 ± 0.07 | 0.90 ± 0.01 |
| | PBP2 | 0.14 ± 0.02 | 0.00 ± 0.16 | 0.06 ± 0.06 | 0.00 ± 0.00 | 0.11 ± 0.04 | 0.18 ± 0.12 |
| | PBP3 | 0.60 ± 0.04 | 0.52 ± 0.05 | 0.48 ± 0.04 | 0.55 ± 0.04 | 0.64 ± 0.09 | 0.61 ± 0.04 |
| | PBP4 | 0.38 ± 0.00 | 0.32 ± 0.25 | NA | 0.24 ± 0.05 | 0.68 ± 0.00 | 0.99 ± 0.01 |
| | PBP5/6 | 0.22 ± 0.03 | 0.09 ± 0.12 | 0.11 ± 0.15 | 0.00 ± 0.06 | 0.22 ± 0.17 | 0.23 ± 0.12 |
| AVI | MIC (mg/L) | >256 | >256 | >256 | >256 | 256 | 256 |
| | PBP1a | 0.00 ± 0.03 | 0.00 ± 0.06 | 0.00 ± 0.01 | 0.00 ± 0.20 | 0.00 ± 0.06 | 0.00 ± 0.13 |
| | PBP1b | 0.05 ± 0.03 | 0.00 ± 0.07 | 0.07 ± 0.01 | 0.00 ± 0.18 | 0.17 ± 0.02 | 0.15 ± 0.16 |
| | PBP2 | 0.00 ± 0.03 | 0.00 ± 0.06 | 0.04 ± 0.01 | 0.00 ± 0.16 | 0.00 ± 0.04 | 0.02 ± 0.18 |
| | PBP3 | 0.00 ± 0.02 | 0.00 ± 0.16 | 0.07 ± 0.02 | 0.00 ± 0.20 | 0.00 ± 0.05 | 0.00 ± 0.09 |
| | PBP4 | 0.18 ± 0.03 | 0.08 ± 0.27 | NA | 0.14 ± 0.19 | 0.53 ± 0.06 | 0.56 ± 0.10 |
| | PBP5/6 | 0.03 ± 0.03 | 0.08 ± 0.11 | 0.00 ± 0.12 | 0.12 ± 0.22 | 0.00 ± 0.12 | 0.07 ± 0.05 |
| REL | MIC (mg/L) | >256 | >256 | >256 | >256 | >256 | >256 |
| | PBP1a | 0.08 ± 0.03 | 0.07 ± 0.08 | 0.00 ± 0.06 | 0.00 ± 0.05 | 0.08 ± 0.04 | 0.17 ± 0.02 |
| | PBP1b | 0.09 ± 0.02 | 0.10 ± 0.16 | 0.00 ± 0.03 | 0.00 ± 0.13 | 0.03 ± 0.10 | 0.22 ± 0.02 |
| | PBP2 | 0.11 ± 0.04 | 0.14 ± 0.21 | 0.00 ± 0.13 | 0.00 ± 0.08 | 0.05 ± 0.14 | 0.25 ± 0.13 |
| | PBP3 | 0.09 ± 0.03 | 0.07 ± 0.22 | 0.00 ± 0.06 | 0.00 ± 0.05 | 0.09 ± 0.05 | 0.16 ± 0.05 |
| | PBP4 | 0.18 ± 0.04 | 0.23 ± 0.25 | NA | 0.05 ± 0.01 | 0.29 ± 0.05 | 0.30 ± 0.03 |
| | PBP5/6 | 0.13 ± 0.04 | 0.19 ± 0.12 | 0.00 ± 0.19 | 0.00 ± 0.01 | 0.08 ± 0.07 | 0.20 ± 0.02 |
| SUL | MIC (mg/L) | >256 | >256 | >256 | >256 | 128 | 128 |
| | PBP1a | 0.00 ± 0.01 | 0.05 ± 0.09 | 0.04 ± 0.05 | 0.10 ± 0.13 | 0.02 ± 0.10 | 0.09 ± 0.11 |
| | PBP1b | 0.00 ± 0.02 | 0.10 ± 0.13 | 0.01 ± 0.05 | 0.11 ± 0.16 | 0.29 ± 0.06 | 0.28 ± 0.10 |
| | PBP2 | 0.00 ± 0.03 | 0.10 ± 0.11 | 0.04 ± 0.15 | 0.10 ± 0.19 | 0.00 ± 0.21 | 0.10 ± 0.21 |
| | PBP3 | 0.12 ± 0.02 | 0.05 ± 0.13 | 0.12 ± 0.12 | 0.04 ± 0.20 | 0.06 ± 0.18 | 0.00 ± 0.15 |
| | PBP4 | 0.00 ± 0.03 | 0.21 ± 0.10 | NA | 0.26 ± 0.16 | 0.43 ± 0.08 | 0.46 ± 0.15 |
| | PBP5/6 | 0.00 ± 0.04 | 0.16 ± 0.17 | 0.01 ± 0.05 | 0.15 ± 0.06 | 0.21 ± 0.09 | 0.19 ± 0.23 |
| TZB | MIC (mg/L) | >256 | >256 | >256 | >256 | 128 | 128 |
| | PBP1a | 0.03 ± 0.03 | 0.00 ± 0.04 | 0.01 ± 0.07 | 0.08 ± 0.05 | 0.00 ± 0.06 | 0.22 ± 0.07 |
| | PBP1b | 0.02 ± 0.02 | 0.00 ± 0.07 | 0.00 ± 0.20 | 0.07 ± 0.09 | 0.13 ± 0.02 | 0.42 ± 0.04 |
| | PBP2 | 0.01 ± 0.04 | 0.00 ± 0.03 | 0.00 ± 0.54 | 0.07 ± 0.13 | 0.00 ± 0.14 | 0.29 ± 0.15 |
| | PBP3 | 0.00 ± 0.04 | 0.00 ± 0.15 | 0.08 ± 0.18 | 0.01 ± 0.08 | 0.00 ± 0.13 | 0.22 ± 0.05 |
| | PBP4 | 0.03 ± 0.01 | 0.00 ± 0.08 | NA | 0.00 ±0.19 | 0.04 ± 0.15 | 0.53 ± 0.13 |
| | PBP5/6 | 0.03 ± 0.04 | 0.00 ± 0.04 | 0.00 ± 0.09 | 0.02 ± 0.13 | 0.00 ± 0.25 | 0.23 ± 0.13 |

[a]Whole-cell-bound PBP fraction for PAO1, OprD porin, PBP4, AmpC β-lactamase, OprM efflux component, and double AmpC and OprM isogenic knockout mutants in the presence of 1/2× PAO1 MIC of each β-lactam. The BLIs AVI, REL, SUL, and TZB were used at a concentration of 4 mg/L. The median values from 3 experiments are shown. Please see Table 2 for drug abbreviations. Data represent mean ± SD unless otherwise indicated.

and the double AmpC-OprM mutant was selective for PBP2. However, amdinocillin MIC reduction (64-fold) was not correlated with the lower impact on PBP occupancy in any of the mutants. AmpC overexpression (Δ*dacB*) caused a 2-fold reduction in aztreonam PBP3 binding, which translated into a 4-fold MIC increase, whereas the reduction of target binding for amdinocillin was mostly noticeable for PBP2.

Penicillin PBP occupancies were significantly affected by OprM inactivation (alone and together with *ampC* inactivation), eliciting up to 12- and 7-fold of increased PBP1a and PBP1b binding, respectively. Subsequent 32- to 64-fold MIC reductions were observed for all the drugs in this class. Inactivation of *dacB* produced a 2- to 4-fold MIC increase for all penicillins, with a reduction (1.2- to 2-fold) in PBP1a, PBP1b, and PBP3 binding.

Finally, a significant increase (1- to 3-fold) of target attainment was observed in the OprM and double AmpC-OprM mutants for the BLIs avibactam, relebactam, sulbactam, and tazobactam. However, only sulbactam and tazobactam displayed a noticeable ~2-fold MIC reduction.

## DISCUSSION

With the current expansion of *P. aeruginosa*-resistant isolates and MDR/XDR high-risk clones continuously evolving to become resistant to the most recently approved drugs, few to no treatment options remain available (2, 3). β-Lactams, proven to have one of the best efficacy and safety profiles, have largely been the cornerstone of antimicrobial therapy against Gram-negative isolates (4). Steady development of novel β-lactams has allowed combinations of two different compounds that are not affected by the same resistance determinants. Furthermore, novel BLIs have been extensively used to either protect the partner β-lactam from β-lactamase hydrolysis or to enhance its activity via complementary target inactivation (24, 25). However, therapeutic failures have been frequently reported with β-lactam combinations, leading to nonoptimal dosage regimens and emergence of resistance (6, 26).

To design and optimize effective β-lactam-based combination therapies, a systematic approach studying the target site penetration and binding is needed. However, as pointed out by several authors, *P. aeruginosa* OM permeability cannot be accurately modeled by penetration-based assays using a β-lactamase as a surrogate (18, 19, 23, 27, 28). To gain insights on the correlations between MICs and periplasmic drug uptake and target binding, we characterized the intact cell PBP occupancy patterns of 11 clinically relevant β-lactams and 4 β-lactamase inhibitors in the wild-type strain PAO1. We determined the contribution of the intrinsic and acquired β-lactam resistance determinants to the PBP occupancy and, thus, β-lactam efficacy in *P. aeruginosa* (e.g., porin expression, efflux, or β-lactamase hydrolysis).

Our whole-cell assay studies PBP binding in the presence of all PBPs at their natural relative abundance in the periplasmic space of intact bacteria, and they are affected by the net influx rate of each drug (i.e., the sum of the effects of porins, efflux pumps, and β-lactamase hydrolysis). Carbapenems required low concentrations to bind to virtually all PBPs in intact PAO1 cells, especially PBP2 and PBP4, as previously described in *E. coli* (10). Imipenem showed the most extensive binding, followed by doripenem and meropenem. Regardless of sharing almost identical PBP-binding values in isolated membrane assays, the considerably faster initial killing rate that characterizes imipenem could be linked with a rather much faster penetration through the OM (29).

However, ertapenem, meropenem, ceftazidime, cefepime, aztreonam, piperacillin, and ticarcillin displayed a moderate intact cell target attainment, although at higher concentrations than that previously reported for *E. coli* (10, 21). In contrast, amdinocillin, carbenicillin, and BLIs scarcely bound selected targets regardless of the higher concentrations used compared to the clinically achievable ones (30, 31). Despite quantitative binding differences, the PCA of the log-transformed whole-cell-bound PBP fraction clustered all the compounds according to their primary target (PBP2 and PBP4 versus PBP3), in agreement with previous studies in *P. aeruginosa* lysed and *E. coli* intact cell assays (6, 10, 31).

A notable strength of our PAO1 assay is that it accounts for the net influx rate, a direct consequence of drug penetration rate, and the impact of the intrinsic β-lactam resistance mechanisms. We wanted to assess if our approach was, in fact, reporting differences in periplasmic accumulation and binding, so the next step was to study the contribution of the major intrinsic and acquired β-lactam resistance mechanisms to PBP target site penetration and binding.

Inactivation of OprM markedly affected meropenem-PBP binding, yet the efflux component had a lower impact on ertapenem and doripenem. Resistance to these compounds is

typically associated with the upregulation of efflux systems, mainly MexAB-OprM (5, 32). Due to its strongly charged hydrophilic side chains, imipenem PBP occupancies remained unaltered (33, 34). MexAB-OprM system is constitutively expressed in almost all *P. aeruginosa* strains, and the substrates for this pump, piperacillin, carbenicillin, ticarcillin, ceftazidime, cefepime, and aztreonam showed the most significant changes in PBP binding and MIC. We were able to assess significant binding increase for the selective targets upon OprM inactivation for amdinocillin (PBP2), penicillins, aztreonam, and cephalosporins (PBP1a, PBP1b, and PBP3), especially cefepime. The differences observed for the latter are a direct consequence of its lower periplasmic uptake associated with MexAB- and MexXY-OprM efflux pumps (35).

One of the limitations of our assay is that MexXY-OprM, which shares the same outer membrane efflux pump channel (OprM), may contribute to selected $\beta$-lactam resistance along with the MexCD-OprJ system. However, these efflux pumps are not typically expressed under normal growth conditions and thus do not mainly contribute to the intrinsic resistance to $\beta$-lactams (36). According to previously reported data, OprD inactivation caused the most notable reduction in target site penetration and binding of imipenem. Doripenem, meropenem, and, especially, ertapenem were less affected (37, 38).

Although AmpC expression is highly inducible in the presence of certain $\beta$-lactams such as imipenem or cefoxitin, it is produced at very low basal levels in wild-type strains (28). The hypersensitivity observed for imipenem has been suggested to be caused by the disruption of *ampC* induction (30, 39, 40). However, it is not clear whether the previously observed increase in *ampC* transcription from reverse transcription-quantitative PCR (qRT-PCR)-based induction assays directly turns into AmpC-specific activity. Nevertheless, in the present study, we were not able to observe a significant increase of PBP binding upon *ampC* inactivation, possibly due to the out-of-phase gene transcription and $\beta$-lactamase production. On the other hand, AmpC constitutive overexpression ($\Delta dacB$) elicited a substantial PBP occupancy reduction for the widely used antipseudomonal penicillins and ceftazidime (PBP1a, PBP1b, and PBP3) and to a lesser extent for the fourth-generation cephalosporin cefepime (PBP1a and PBP1b) and aztreonam (PBP3) (41). No significant differences were observed for any of the carbapenems, in consensus with the MIC shift, as they are not good AmpC substrates (39, 42, 43).

As previously described, novel BLIs did not show either an extensive PBP occupancy pattern or a significant increase or decrease of their primary target attainment in any of the studied strains. Thus, our observations are in accordance with previous works describing that the positive effects of new BLIs are mostly preserved on strains that simultaneously overproduce AmpC and MexAB-OprM (30, 44). However, they would benefit from efflux inhibition or outer membrane permeabilization.

Our work presents the first PBP occupancy data set in intact *P. aeruginosa* (including the most relevant $\beta$-lactam resistance mechanisms), a step forward to enable the selection of $\beta$-lactams and combinations for future synergy studies with $\beta$-lactamase inhibitors and other antibiotic classes. Our study has some potential weaknesses, such as the higher variability in PBP binding for low-affinity targets and the presence of multiple RND efflux systems. Nevertheless, it is our honest belief that this approach is a step forward to understanding $\beta$-lactam resistance, mechanism of action, and efficacy, generating valuable insights for quantitative and systems pharmacology (QSP) predictive models and translational drug development (29). Future studies elucidating $\beta$-lactamase-driven kinetic target binding, intact cell drug-drug interactions, the time course of PBP inactivation and resynthesis, and the extent of individual PBP saturation required for bacterial growth inhibition in single cells will greatly enhance the understanding of $\beta$-lactam target site binding and action.

## MATERIALS AND METHODS

**Bacterial strains, *in vitro* susceptibility testing, and time-kill curves.** The MIC of $\beta$-lactams and $\beta$-lactamase inhibitors was determined by the standard Clinical and Laboratory Standards Institute (CLSI) broth microdilution method (45) for wild-type reference strain *P. aeruginosa* PAO1 and OprD porin (PA$\Delta$oprD), PBP4 (PA$\Delta$dacB), AmpC $\beta$-lactamase (PA$\Delta$ampC), OprM efflux component (PA$\Delta$oprM), and double AmpC and OprM (PA$\Delta$ampCoprM) previously constructed isogenic knockout mutants (39, 46). MIC values were determined from at least two independent experiments. Imipenem was purchased from Fresenius Kabi

**TABLE 2** MICs for all studied strains

| Drug[b] | MIC (mg/L) of strain[a]: | | | | | |
|---|---|---|---|---|---|---|
| | PAO1 | ΔoprD | ΔdacB | ΔampC | ΔoprM | ΔampCoprM |
| IPM | 1 | 8 | 1 | 0.125 | 1 | 0.25 |
| DOR | 0.5 | 1 | 0.5 | 0.125 | 0.125 | 0.06 |
| MEM | 0.5 | 1 | 1 | 0.125 | 0.032 | 0.032 |
| ETP | 8 | 16 | 16 | 4 | 2 | 0.5 |
| CAZ | 1 | 1 | 8 | 0.5 | 0.25 | 0.25 |
| FEP | 1 | 1 | 4 | 1 | 0.125 | 0.125 |
| ATM | 2 | 2 | 8 | 2 | 0.125 | 0.125 |
| MEC | 256 | 256 | >256 | 256 | 4 | 4 |
| CAR | 64 | 64 | 128 | 64 | 1 | 1 |
| PIP | 4 | 4 | 16 | 2 | 0.25 | 0.125 |
| TIC | 16 | 16 | 64 | 16 | 0.5 | 0.5 |
| AVI | >256 | >256 | >256 | >256 | 256 | 256 |
| REL | >256 | >256 | >256 | >256 | >256 | >256 |
| SUL | >256 | >256 | >256 | >256 | 128 | 128 |
| TZB | >256 | >256 | >256 | >256 | 128 | 128 |

[a]OprD porin, AmpC β-lactamase, PBP4, OprM efflux component, and double AmpC-OprM isogenic PAO1 knockout mutants.

[b]IPM, imipenem; DOR, doripenem; MEM, meropenem; ETP, ertapenem; CAZ, ceftazidime; FEP, cefepime; ATM, aztreonam; MEC, mecillinam; CAR, carbenicillin; PIP, piperacillin; TIC, ticarcillin; AVI, avibactam; REL, relebactam; SUL, sulbactam and TZB, tazobactam. The median values from 3 experiments are shown.

(Spain); meropenem from Aurovitas (Spain); ertapenem from Merck Sharp & Dohme (Netherlands); ceftazidime from Laboratorios Normon (Spain); cefepime from Accord Healthcare (Spain); aztreonam from Bristol-Myers Squibb (Spain); and doripenem, amdinocillin, carbenicillin, piperacillin, ticarcillin, avibactam, relebactam, sulbactam, and tazobactam from MedChemExpress (Sweden). To monitor cell integrity throughout the assay, static concentration time-kill curves (SCTK) were determined. PAO1 and the 5 isogenic mutants were grown overnight in cation-adjusted Mueller-Hinton broth (CAMHB) and were inoculated with compound (starting inoculum of 6.5 $\log_{10}$ CFU/mL) into 50-mL conical tubes containing wild-type 1/2× PAO1 MIC of each compound. Serial dilutions were plated after 15, 30, and 60 min of incubation. MIC values and SCTK were determined from at least three independent experiments.

**Whole-cell PBP binding assay.** Mid-exponential growing *P. aeruginosa* PAO1 cultures (6.5 $\log_{10}$ CFU/mL) were incubated in CAMHB at 37°C (180 rpm) containing 1/2× PAO1 MIC (Table 2) of each compound (except β-lactamase inhibitors, used at a fixed concentration of 4 mg/L, the concentration used in combination with β-lactam partners for sensitivity studies). Untreated and treated samples were taken 30 min postincubation, kept in ice, and centrifuged (3,220 × *g* for 10 min at 4°C). Bacterial pellets were washed in 30 mL of phosphate-buffered saline (PBS) buffer (pH 7.5) four times, resuspended in 10 mL of PBS, and gently sonicated using a digital Sonifier unit model S-450D (Branson Ultrasonics Corp., Danbury, CT) (immersed in an ice bath). Membranes containing antibiotic-bound PBPs were collected by ultracentrifugation (150,000 × *g*, 30 min, 4°C) using an Optima L100XP ultracentrifuge (Beckman Coulter, Inc., Palo Alto, CA) and resuspended in 60 μL of PBS. Total protein content was measured using the Quick Start Bradford protein assay kit with bovine serum albumin as standard (Bio-Rad Laboratories, Hercules, CA), according to the manufacturer's instructions.

To determine the bound PBP fraction, membranes containing PBPs of PAO1 (10 μg) were labeled with 25 μM fluorescent penicillin Bocillin FL (15 min at 37°C) (47). Labeled PBPs were separated through 4 to 15% SDS-polyacrylamide gels (Bio-Rad Laboratories) and visualized using a Typhoon FLA 9500 biomolecular imager (GE Healthcare Bio-Sciences AB, Uppsala, Sweden) (excitation at 488 nm and emission at 530 nm). Binding to different PBPs was determined from at least three independent experiments using ImageQuant TL software v8.1.0.0 (GE Healthcare Bio-Sciences AB). Serial dilutions were plated to assess bacterial growth and killing. To rule out dissociation of the PBP-drug complex during membrane isolation and displacement of the drug, Bocillin FL labeling was performed as well in intact cells, and different incubation times were analyzed. The same approach was used to compare MIC shifts and the differential periplasmic drug accumulation and PBP binding under the same extracellular drug concentrations (1/2× PAO1 MIC) in a series of isogenic PAO1 knockout mutants (PAΔoprD, PAΔdacB, PAΔampC, PAΔoprM, and PAΔampCoprM).

**Data analysis.** GraphPad Prism v7.01 software was used for graphical representation and statistical analysis. Quantitative variables were compared using unpaired Student's *t* test (two independent groups) or one-way analysis of variance (ANOVA) with *post hoc* Tukey's multiple-comparison test (multiple independent groups).

Principal-component analysis (PCA) was performed in XLSTAT software (v2021.5; Addinsoft). We analyzed the logarithmic PBP-binding data of 15 antibiotics in the *P. aeruginosa* PAO1 strain. For the PCA, the Pearson correlation to group variables and observations based on their similarity levels were applied. The distances between the observations were calculated according to the Mahalanobis distance. Bound PBP fraction values were represented on a log scale, and compounds were clustered according to their positions on the first two eigenvectors (which explained most of the variance in PBP binding).

## SUPPLEMENTAL MATERIAL

Supplemental material is available online only.

**SUPPLEMENTAL FILE 1**, PDF file, 0.5 MB.

## ACKNOWLEDGMENTS

B.M. received funds from RADIX17/3-1 fellowship and RADIX17/3-2 grant program within the FUTURMed project IdISBa Research Institute of Health Sciences of the Balearic Islands, Hospital Universitario Son Espases, Palma, Spain, and sustainable tourism tax, Govern de les Illes Balears, by the Miguel Servet Research Contract Program CP20/00138 from the National Institute of Health Carlos III (ISCIII) and by the Agencia Estatal de Investigación (AEI; state research agency), Spain, through the Plan Estatal de Investigación Científica PROYECTOS DE I+D+i PID2020-112654RB-I00. The assay development part of this work was supported by the award R01AI136803 to B.M. from the National Institute of Allergy and Infectious Diseases (NIAID). The content of this paper is solely the responsibility of the authors and does not necessarily represent the official views of the National Institute of Allergy and Infectious Diseases, the National Institutes of Health (NIH).

A.O. received funds from the by the Ministerio de Economía y Competitivad of Spain, Instituto de Salud Carlos II, cofinanced by European Regional Development Fund, "a way to achieve Europe" (ERDF), through the Spanish Network for the Research in Infectious Diseases (RD16/0016).

S.L.-A. received funds from FOLIUM 17/12 fellowship program within the FUTURMed project from the Fundación Instituto de Investigación Sanitaria Illes Balears (financed by the 2017 annual plan of the sustainable tourism tax and at 50% with charge to the ESF Operational Program 2014 to 2020 of the Balearic Islands).

A.O. has received fees as speaker and/or research grants from MSD, Pfizer, and Wockhardt. All other authors have no conflict of interest to declare.

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
