## [Reviewer comments · Microbiology Spectrum]

Microbiology Spectrum

PBP target profiling by β -lactam and β -lactamase inhibitors in intact *Pseudomonas aeruginosa*: Effects of the intrinsic and acquired resistance determinants on the periplasmic drug availability

Maria Montaner, Silvia Lopez-Argüello, Antonio Oliver, and Bartolome Moya

Corresponding Author(s): Bartolome Moya, Fundacio Institut d'Investigacio Sanitaria Illes Balears

Review Timeline:

Submission Date:	August 4, 2022
Editorial Decision:	October 5, 2022
Revision Received:	November 1, 2022
Accepted:	November 21, 2022

Editor: Krisztina Papp-Wallace

Reviewer(s): The reviewers have opted to remain anonymous.

Transaction Report:

DOI: <https://doi.org/10.1128/spectrum.03038-22>

October 5, 2022

Dr. Bartolome Moya
Health Research Institute of the Balearic Islands (IdISBa)
Microbiology
Carretera de Valldemossa, 79
Crta. Valldemossa, 79
Palma, Balearic Islands 07120
Spain

Re: Spectrum03038-22 (PBP target profiling by β -lactam and β -lactamase inhibitors in intact *Pseudomonas aeruginosa*: Effects of the intrinsic and acquired resistance determinants on the periplasmic drug availability)

Dear Dr. Bartolome Moya:

Link Not Available

Sincerely,

Krisztina Papp-Wallace

Journals Department
Reviewer comments:

Reviewer #1 (Comments for the Author):

The manuscript investigates the binding of beta-lactams and beta-lactam inhibitors to periplasmic PBPs in *P. aeruginosa* PAO1 WT and knockout strains impacting various drug resistance mechanisms (porin, beta-lactamase, and efflux). The results offer insightful new information for understanding PBP interactions with different beta-lactams, including the influences of the most common drug resistance mechanisms against these antibiotics. The concise manuscript is well written. Some minor revisions are needed.

- 1) The amount of PBP bound by a particular beta-lactam was determined by the remaining PBP molecules available to react with Bocillin FL, in comparison to those bound by Bocillin FL in the absence of the beta-lactam. While the authors presented the gels showing the PBPs bound by Bocillin, there did not seem to be a control experiment demonstrating that the use of beta-lactam does not change the amount of total protein for a particular PBP.
- 2) AmpC hyperexpression caused noticeable MIC increases for some beta-lactams but not others. Does this pattern agree with the different activities of AmpC on these beta-lactams in previous biochemical studies?
- 3) Although all these PBPs bind to beta-lactams, their contributions to *P. aeruginosa* survival vary. Do the results in this paper suggest any PBPs are more valuable antibiotic targets than others? Do they agree with previous studies on these PBPs, including both the in vitro IC50s using purified proteins (if available) and whether they are essential?
- 4) Line 88, does 'the same PBP-binding affinities (IC50)' apply to one PBP or against all PBPs from the same bacteria? As the authors have shown, there are many PBPs that bind to each beta-lactam usually with varied affinities.
- 5) Table 1, footnote a, the listing of 'PBP4' in the knockout strains is out of order compared with the columns in the table.

Reviewer #3 (Comments for the Author):

The authors compare inhibition of binding of a fluorescent penicillin to *P. aeruginosa* PBPs by a set of representative β -lactam antibiotics and β -lactamase inhibitors. The experimental design unfortunately limits the conclusions that can be drawn and some revision is required to take these limitations into account.

(1) The decision to use exposure at a fixed (half-MIC) concentration and time point makes interpretation of the various observed levels of PBP inhibition difficult. Since the rate of influx and the rate of reaction with PBP at low concentration are directly proportional to inhibitor concentration, performing the experiment at concentrations that differ by as much as 500 fold (e.g. meropenem and mecillinam) means that the reactions in very different conditions are being compared and therefore conclusions about rate of entry are very uncertain. Possibly, as performed, the experiments tell more about levels of occupancy that are necessary to elicit growth inhibition by the different antibiotics.

The experiments would have been better performed with different exposure times and inhibitor concentrations to assess kinetics. In this respect, the experiments with the BLIs, performed at the same concentration, are more interesting. It would be good to use a separate panel in Fig. 1, with a more useful scale, for this data set.

(2) Line 165. Has the Bocillin FL assay been calibrated for *P. aeruginosa* PBPs? Are the chosen concentration and incubation time appropriate to saturate all the PBPs? In the absence of this information it is difficult to interpret the data presented in Fig. 3. It appears that, for a particular PBP, the degree of labelling is similar in the different strains (except $\Delta dacB$) but comparison of levels between PBPs is uncertain. For example, what is the significance of the difference between PBP1b and PBP2 labelling? Are the different levels really due to different amounts of protein or is the labelling of PBP2 less complete because of slower reaction or lower affinity?

(3) Line 204. The assay used does not provide information about affinity: it is only possible to say that the reaction had a greater or a lesser extent in the particular reaction conditions used for each inhibitor and with each PBP. The rates of entry and reaction as well as affinity for target all play a role in the extent of inhibition but separating these requires a more comprehensive approach, as outlined above. Reference to affinity should be deleted throughout.

(4) L. 378. There is no "mechanistic data", in terms of descriptions of rates and affinities, and this statement should be modified.

Staff Comments:

Preparing Revision Guidelines

Please return the manuscript within 60 days; if you cannot complete the modification within this time period, please contact me. If you do not wish to modify the manuscript and prefer to submit it to another journal, please notify me of your decision immediately so that the manuscript may be formally withdrawn from consideration by Microbiology Spectrum.

Reviewer comments:

Reviewer #1 (Comments for the Author):

The manuscript investigates the binding of beta-lactams and beta-lactam inhibitors to periplasmic PBPs in *P. aeruginosa* PAO1 WT and knockout strains impacting various drug resistance mechanisms (porin, beta-lactamase, and efflux). The results offer insightful new information for understanding PBP interactions with different beta-lactams, including the influences of the most common drug resistance mechanisms against these antibiotics. The concise manuscript is well written. Some minor revisions are needed.

1) The amount of PBP bound by a particular beta-lactam was determined by the remaining PBP molecules available to react with Bocillin FL, in comparison to those bound by Bocillin FL in the absence of the beta-lactam. While the authors presented the gels showing the PBPs bound by Bocillin, there did not seem to be a control experiment demonstrating that the use of beta-lactam does not change the amount of total protein for a particular PBP.

In this manuscript we wanted to highlight the PBP occupancy of the different β -lactams and BLIs in intact cells, and how the different resistance mechanisms interfere with the overall target binding balance. However in another manuscript that is currently in preparation we have measured the differential expression of each PBP under β -lactam exposure via qRT-PCR and we haven't found significant differences. As opposed to PBP induced expression after exposure to sub-MIC concentrations observed in other bacterial species, for example *S. aureus*, *P. aeruginosa* PBP expression appears to be dependent on the growth phase and not significantly altered upon antibiotic exposure.

2) AmpC hyperexpression caused noticeable MIC increases for some beta-lactams but not others. Does this pattern agree with the different activities of AmpC on these beta-lactams in previous biochemical studies?

The results from our present study are in agreement with previous studies, according to the MIC changes observed in our work and the β -lactamase activity studied in previous works (now referenced in the main text). PBP

occupancy reduction for the antipseudomonal penicillins and ceftazidime (PBP1a, 1b and 3), cefepime (PBP1a and 1b) and aztreonam (PBP3) correlated with MIC changes and previous observations as specific β -lactamase activity and kinetic hydrolysis profile measurements.

3) Although all these PBPs bind to beta-lactams, their contributions to *P. aeruginosa* survival vary. Do the results in this paper suggest any PBPs are more valuable antibiotic targets than others? Do they agree with previous studies on these PBPs, including both the in vitro IC50s using purified proteins (if available) and whether they are essential?

The results obtained in this paper, confirm what has already been observed in previous publications. PBP1a, 1b, PBP2 and PBP3 are considered to be essential for *P. aeruginosa* growth and survival. Most extensive killing is obtained with the inhibition of all High Molecular Weight essential PBPs: PBP1a, 1b, 2 and 3 elicited by carbapenems, especially imipenem (fast penetration). And different degrees of inhibition and combinations within these PBP yield different rates and extent of bacterial killing. Several publications (Legaree et al, JAC 2007, Chen et al, AAC 2017) have postulated PBP3 as the most important and essential receptor for *P. aeruginosa* survival. In agreement with preliminary results obtained in our lab, that remains true when a low to mid bacterial inoculum is tested (10EXP5 – 10EXP7); however, when bacterial inoculum exceeds 10EXP9, the key target for bacterial killing falls on PBP2. Our IC50 values are in accordance with previously reported values in the literature for the available drugs.

4) Line 88, does 'the same PBP-binding affinities (IC50)' apply to one PBP or against all PBPs from the same bacteria? As the authors have shown, there are many PBPs that bind to each beta-lactam usually with varied affinities.

We refer to PBP binding affinities taking in account all the receptors, i.e., drugs that possess the same PBP binding profile and PBP selectivity (affinity for a defined group of PBPs). The sentence has been modified in the manuscript.

5) Table 1, footnote a, the listing of 'PBP4' in the knockout strains is out of order compared with the columns in the table.

The order has been changed accordingly.

Reviewer #3 (Comments for the Author):

The authors compare inhibition of binding of a fluorescent penicillin to *P. aeruginosa* PBPs by a set of representative β -lactam antibiotics and β -lactamase inhibitors. The experimental design unfortunately limits the conclusions that can be drawn and some revision is required to take these limitations into account.

(1) The decision to use exposure at a fixed (half-MIC) concentration and time point makes interpretation of the various observed levels of PBP inhibition difficult. Since the rate of influx and the rate of reaction with PBP at low concentration are directly proportional to inhibitor concentration, performing the experiment at concentrations that differ by as much as 500 fold (e.g. meropenem and mecillinam) means that the reactions in very different conditions are being compared and therefore conclusions about rate of entry are very uncertain. Possibly, as performed, the experiments tell more about levels of occupancy that are necessary to elicit growth inhibition by the different antibiotics.

The experiments would have been better performed with different exposure times and inhibitor concentrations to assess kinetics. In this respect, the experiments with the BLIs, performed at the same concentration, are more interesting. It would be good to use a separate panel in Fig. 1, with a more useful scale, for this data set.

We agree with the reviewer, and we have modified the text accordingly. We decided to use 1/2 MIC concentration of all the studied drugs to relate PBP occupancy levels with the extracellular drug concentration, and afterwards to assess if the differences in MIC of the intrinsic or acquired resistance mechanisms could be related to changes in PBP occupancy. As this procedure is very time consuming, our priority was to determine the feasibility of the method and to assess if it would be sensitive to observe differences in target binding in the presence or absence of resistance determinants. Our next step will be to improve the method to for high throughput screening and to determine the binding kinetics.

We have changed the scale on the BLI panel in Fig. 1 as suggested by the reviewer.

(2) Line 165. Has the Bocillin FL assay been calibrated for *P. aeruginosa* PBPs? Are the chosen concentration and incubation time appropriate to saturate all the PBPs? In the absence of this information it is difficult to interpret the data presented in Fig. 3. It appears that, for a particular PBP, the degree of labelling is similar in the different strains (except $\Delta dacB$) but comparison of levels between PBPs is uncertain. For example, what is the significance of the difference between PBP1b and PBP2 labelling? Are the different levels really due to different amounts of protein or is the labelling of PBP2 less complete because of slower reaction or lower affinity?

While we certainly didn't reach steady-state conditions due to much higher enzyme concentrations than the substrate (Bocillin FL). Our assay has been previously calibrated for incubation time (5, 10, 15, 30, 90 mins), protein concentration (0.2, 0.5, 0.75, 1, 2, 4 mg/mL) and bocillin concentration (2.5, 5, 10, 15, 25 μ M). In all the conditions tested, the differences observed for band intensity of PBP1b and 2, as an example, were always constant. We are aware of the substantially lower PBP2 acylation rate constant compared with PBP3 as Shapiro et al. described for their soluble forms (Analytical Biochemistry 2013 and 2014). Direct quantitation of individual PBP/cell has been performed in *E. coli* initially by Spratt Eur. J. Biochem, 1977 and afterwards by Dougherty et al. Journal of Bacteriology 1996, however no such information is available for *P. aeruginosa*. Given the limitations of the assay we wanted to highlight that band signals were not altered in the different isogenic mutants; it was not our intention to compare band intensities of different PBPs for a given strain. We have changed the text in the manuscript to highlight our goal.

(3) Line 204. The assay used does not provide information about affinity: it is only possible to say that the reaction had a greater or a lesser extent in the particular reaction conditions used for each inhibitor and with each PBP. The rates of entry and reaction as well as affinity for target all play a role in the extent of inhibition but separating these requires a more comprehensive approach, as outlined above. Reference to affinity should be deleted throughout.

We agree with the reviewer comments, and we have changed the text accordingly.

(4) L. 378. There is no "mechanistic data", in terms of descriptions of rates and affinities, and this statement should be modified.

The statement has been modified accordingly.

November 21, 2022

Dr. Bartolome Moya
Fundacio Institut d'Investigacio Sanitaria Illes Balears
Microbiology
Carretera de Valldemossa, 79
Crta. Valldemossa, 79
Palma, Balearic Islands 07120
Spain

Re: Spectrum03038-22R1 (PBP target profiling by β -lactam and β -lactamase inhibitors in intact *Pseudomonas aeruginosa*: Effects of the intrinsic and acquired resistance determinants on the periplasmic drug availability)

Dear Dr. Bartolome Moya:

Your manuscript has been accepted, and I am forwarding it to the ASM Journals Department for publication. You will be notified when your proofs are ready to be viewed.

Sincerely,

Krisztina Papp-Wallace
Editor, Microbiology Spectrum
